# The effects of type of recovery in resistance exercise on responses of platelet indices and hemodynamic variables

**Mohammad Soltani**[1]*, **Atefe Sarvestan**[2◉], **Fatemeh Hoseinzadeh**[2◉], **Sajad Ahmadizad**[2], **J. Derek Kingsley**[3]*

**1** Centre for Heart, Lung and Vascular Health, School of Health and Exercise Sciences, University of British Columbia, Kelowna, Canada, **2** Faculty of Sport Sciences and Health, Department of Biological Sciences in Sport, Shahid Beheshti University, Tehran, Iran, **3** Exercise Science and Exercise Physiology, Kent State University, Kent, OH, United States of America

◉ These authors contributed equally to this work.
* jkingsle@kent.edu (JDK); moha.soltani@ubc.ca (MS)

**Editor:** Daniel Boullosa, Universidad de León Facultad de la Ciencias de la Actividad Física y el Deporte: Universidad de Leon Facultad de la Ciencias de la Actividad Fisica y el Deporte, SPAIN

## Abstract

To examine the effects of two different volume-matched resistance exercise (RE) recovery protocols (passive and active) on platelet indices and hemodynamic variables. Twelve Healthy participants (mean ± SD; 25 ± 3 yrs) completed a traditional resistance exercise (TRE) protocol that included three sets of six repetitions at 80% one repetition maximum (1RM) with two minutes passive recovery between sets, exercises and an interval resistance exercise (IRE) protocol that included three sets of six repetitions at 60%1RM followed by active recovery including six repetitions of the same exercise at 20%1RM. Blood samples for multiple platelet indices were taken before the protocols, immediately-post (IP), and after 1-hour recovery. Hemodynamic variables were measured before, IP, and every five minutes during recovery. Mean platelet volume and platelet large cell ratio P_LCR decreased from baseline to recovery. Heart rate (HR) and rate pressure product (RPP) were augmented at IP following IRE compared to TRE. HR was significantly elevated for 20 minutes after both RE protocols, and RPP recovered by five minutes. Systolic blood pressure was increased at IP compared to baseline and all recovery time points for both RE protocols. Our research demonstrated that both RE protocols, produced transient increases in platelet indices (MPV, and P_LCR) and hemodynamic variables (SBP, HR, and RPP), all of which returned to baseline within an hour. Notably, the IRE protocol elicited a greater increase in HR and RPP compared to the TRE protocol.

## Introduction

Cardiovascular disease is the top cause of mortality and morbidity in the world [1]. Deteriorated platelet function, platelet hyperactivity, and impaired hemodynamic properties are important variables that play a vital role in the development and progression of cardiovascular diseases [2]. Furthermore, data have demonstrated that disturbances of platelet indices and hemodynamic variables are associated with cardiovascular events and even mortality [3,4].

**Data Availability Statement:** All relevant data are within the paper and its Supporting Information files.

**Funding:** The authors received no specific funding for this work.

**Competing interests:** The authors have declared that no competing interests exist.

Interestingly, studies have revealed acute exercise, particularly of high intensity, triggers a series of physiological responses. These include an elevation in catecholamines, intensified spleen contraction, an increase in fibrinogen levels, a reduction in nitric oxide (NO) bioavailability, and an upsurge in plasma p-selectin levels. Collectively, these changes contribute to enhanced platelet hyperaggregability and impaired platelet function. These alterations could be associated with exercise-related cardiac events and increase the risk of sudden cardiac death two to three-fold [5–12]. While these data investigate high-intensity exercise specifically, other forms of exercise, such as resistance exercise (RE), are needed to further investigate the effects of exercise on platelet indices and hemodynamic variables.

Resistance exercise training has recently attracted much attention in athletes as well as the general population, as a means to increase muscle mass, physical function, and reduce cardiovascular disease risk factors [13]. Acutely, it has been demonstrated that RE affects not only platelet activity and function but also hemodynamic variables [14,15]. Ahmadizad et al. (2006) showed that RE increased platelet activation indicated by an increase in platelet aggregation and an elevation in beta-thromboglobulin (B-TG) [14]. However, this increase was transient and returned to the baseline level after 30-minutes of recovery [14]. Furthermore, these same authors also reported in a separate study that different RE intensities (40%, 60%, and 80% of one-repetition maximum (1RM)) were associated with increased platelet count (PLT), plateletcrit (PCT), and mean platelet volume (MPV), but platelet activation (as shown by B-TG) increased immediately after the RE only in the protocol with the higher intensity (80%1RM) [16]. Moreover, acute RE, when performed with high load (>80%1RM), has also been associated with deteriorating hemodynamic variables and elevated blood pressure (BP) more than aerobic and low-load RE (<50%1RM), and thus, RE could acutely increase the risk of cardiovascular complications [17–19]. Intensity, volume, rest periods between sets, and frequency are favorable variables in designing RE that have an effective role in stimulating thrombocytosis, platelet activity, and may be manipulated to limit the deterioration of hemodynamic variables [14,20,21]. Although the intensity of exercise, specifically in RE, might play a role in platelet activation and hemodynamic properties [9,16], it is not currently understood how recovery between the RE exercises can be an influential variable on platelet indices and hemodynamic variables. Therefore, the purpose of the present study was to examine the effects of RE with different types of recovery (active and passive) by using a traditional resistance exercise (TRE) protocol and an interval resistance exercise (IRE) protocol on platelet indices and hemodynamic variables. We proposed the hypothesis that the IRE protocol, due to its heightened mechanical (increased muscle contraction) and metabolic stimulation (elevated levels of H+, PCO2, lactate, bicarbonate, potassium, sodium, etc.), will trigger a greater physiological stress response. This response, driven by the activation of exercise pressor reflex mechanisms, specifically mechano- and metaboreflexes, is anticipated to be more prominent than the TRE protocol. Consequently, we expected more pronounced and an immediate increase in platelet indices and hemodynamic variables following the implementation of the IRE protocol as compared to the TRE protocol. Our second hypothesis was that recovery from the IRE protocol would result in a quicker return of the platelet activity and hemodynamics to baseline compared to the TRE protocol.

## Methods

Twelve healthy men (n: 6) and women (n: 6) participated in this study. The characteristics of the participants are presented in Table 1. The study inclusion criteria included: age between 20 to 30 years, nonsmoker, no consumption of alcohol in the last month, and had no experiences or diagnosed illness (Cardiovascular disease, hypertension, diabetes, etc.). Participants who used dietary supplements or medication that could affect platelet function or hemodynamics

**Table 1. Participant characteristics (Means, SD ±, N = 12).**

| Characteristics | Age (yr.) | Height (cm) | Weight (kg) | BMI |
|---|---|---|---|---|
| Men | 26.9 ± 2.9 | 179 ± 5.8 | 79.6 ± 7.7 | 24.6 ± 1.8 |
| Women | 24.1 ± 0.8 | 165 ± 4.9 | 63.4 ± 6.1 | 23.0 ± 1.7 |

**BMI**, body mass index.

variables were excluded from the study. The current menstrual cycle stage and use of contraceptives were reported by women and in order to minimize the impact of oestrogen, our protocols for female participants were strategically conducted within the first five days of the early follicular phase of their menstrual cycle, a period when oestrogen levels are usually at their nadir. Participants were assessed by a physician, completed the physical questionnaire readiness (PAR-Q) [22], and medical health/history questionnaires in order to evaluate the participants' study eligibility. All participants after receiving a thorough explanation of the study procedures, potential risks and benefits, confidentiality measures, and their right to withdraw from the study at any point without negative consequences completed a written consent form at the beginning of the study which was confirmed by the university ethic committee already. The study procedures retrospectively (Exercise protocols and laboratory measurements) were reviewed and approved by the Research and Ethics Committee of local University (Ethic code: IR.SBU.RCE.1399.003, Ethic Committee Members: Saadollah Nasiri Ghedari, Babak Shokri, Hamidreza Pouretemad, Mohsen Dehghani, Fariborz Hovanloo, Hassan RAJABI-MAHAM, Mohammed Rasekh, Mojtaba Zarei, Abdolmajid Mahdavidamghani, Atousa Aliahmadi). All procedures were performed ethically according to the latest revision of the Declaration of Helsinki. In our study, we adhere strictly to the highest ethical standards to protect participant privacy and confidentiality. To ensure this, all personal identifying information was removed from the dataset during the initial data collection process, before any analysis was conducted. As such, the authors did not have access to any information that could identify individual participants either during or after data collection.

The entire study, including all measurements, procedures, and blood sampling, was conducted from August to October of 2019.

## Inclusivity in global research

Additional information regarding the ethical, cultural, and scientific considerations specific to inclusivity in global research is included in the Supporting Information (SX Checklist).

## Experimental design

Four separate sessions, each separated by one week, were designed to conduct the study. The first session was the familiarization of participants with RE equipment and exercises, and study procedures. In addition, height (Seca, Germany), body weight (Seca, Germany), and BMI (calculated by dividing body weight (kg) by the square of their height ($m^2$)) were measured in this session. In the second session, the participants' 1RM was measured for determining maximal strength. In the third and fourth sessions, participants performed two RE protocols: traditional resistance exercise (TRE) and interval resistance exercise (IRE) in a randomized crossover design. Both RE protocols were performed at the same time of day (9 p.m. to 11 p.m.) and similar environmental conditions (~22°C and ~55% humidity). The participants were asked to refrain from drinking any caffeinated beverages 24 h before experiments and to avoid moderate to vigorous exercise 48 hours before the study's procedures.

## 1RM and RE protocols

After the familiarization session, the 1RM was measured for six exercises (lying Machine Squat, lat pulldown, leg press, shoulder press, chest press, and standing calf raise). For each 1RM, load progressively increased until the participant could not successfully complete the lift. Participants rested for two minutes between exercises and attempts. Five attempts per lift were considered to achieve their 1RM [21]. Both RE protocols began with 10 minutes warmup (General and specific), and then the RE session for both protocols using the listed exercises above in that same order. During the TRE protocol participants performed three sets of six repetitions at 80%1RM for each exercise, associated with two minutes passive recovery between sets and exercises. For the IRE protocol participants completed all exercises with three sets of six repetitions at 60%1RM, followed by 20 second active recovery, (6 repetitions at 20%1RM). After finishing the three sets of a particular exercise, participants were given a two-minute inactive rest period before starting a new exercise. Both sessions included three phases: first, participants were seated for 30 minutes, then baseline measurements (blood sample, hemodynamic variables, and heart rate (HR)) were taken. The second phase included the RE protocols (TRT or IRE), and the third phase consisted of variables taken immediately after RE up to 60 minutes (Hemodynamic variables were measured every 5 minutes, while blood samples for analyzing platelet indices were collected at the 60-minute mark) of seated recovery. After the RE protocols, systolic blood pressure (SBP), diastolic blood pressure (DBP), mean arterial pressure (MAP), and HR were measured every 5 minutes until 60 minutes. Furthermore, rate pressure product (RPP) was calculated by HR and SBP (HR × SBP in mm Hg). In addition, rating of perceived exertion (RPE) was measured immediately after both RE protocols. The details of both protocols are presented in Table 2. Volume and intensity of both protocols were equalized:

Sets x Reps x Weight = Volume-Load
IRE (Exercise load: 60% + rest exercise load 20%) X 3 sets = 180 AU
TRE (Exercise load: 80% + rest exercise load 0%) X 3 sets = 180 AU

## Blood sampling and analysis

Blood samples were collected in EDTA (Ethylenediaminetetraacetic acid, BD Vacutainer 7.2mg K2EDTA, 4.0mL, Becton Dickinson & Company, Plymouth, UK) or sodium citrate tubes (Coagulation Sodium Citrate 3.2%, Greiner Bio-One GmbH, Frickenhausen, Germany). EDTA blood was analyzed for measurement of platelet indices consisted of platelet count

**Table 2. Exercises, sets, repetitions, and rest intervals between exercises and sets and load of exercise for each session of the TRE, and IRE.**

| Sessions | TRE | IRE |
|---|---|---|
| | **Exercises:** lying Machine, Squat, lat pulldown, leg press, shoulder press, chest press, and standing calf raise<br>**Sets:** 3<br>**Repetitions:** 6<br>**Rest interval between exercises:** 2 min<br>**Rest between each set:** 2 min<br>**Load:** 80% of 1RM<br>**Total load:** 80% of 1RM | **Exercises:** lying Machine, Squat, lat pulldown, leg press, shoulder press, chest press, and standing calf raise<br>**Sets:** 3<br>**Repetitions:** 6<br>**Rest interval between exercises:** active rest with 20% of 1RM<br>**Load:** 60% of 1RM<br>**Total load:** 80% 1RM |

**TRE,** Traditional resistance exercise

**IRE,** Interval resistance exercise

**1RM**, One-repetition maximum.

(PLT), platelet large cell ratio (PLC-R), mean platelet volume (MPV), and platelet distribution width (PDW) by using a blood cell counter (XP-300, Sysmex, US). Whole sodium citrate blood (Nine portions of whole blood were added to one portion of trisodium citrate dehydrate) was centrifuged at 2500 g for 10 minutes at 4<C. Then blood plasma was analyzed (Biomeriux option 4 plus, Germany) to the determination of fibrinogen concentration. Plasma volume was identified according to equations described by Dill and Costill by using hematocrit and hemoglobin value before and after RE [23].

$$\Delta PV(\%) = 100 \times ((Hbpre \div Hbpost) \times (100 - Htc) - 1)$$

$$\Delta PV(\%) = 100 \times ((Hbpre \div Hbpost) \times (100 - Htcpost) \div (100 - Hctpre) - 1)$$

where Htc is in % and Hb in g/dL.

## Statistical analysis

Analyses of all data were conducted by Statistical Package for Social Sciences (SPSS) version 22 (IBM SPSS, Armonk, NY, USA). Shapiro-Wilk test was used to determine the normality of the data. Repeated measures of analysis of variance (ANOVA) (2 protocols × 3 times) and (2 protocols × 14 times) were employed for platelet indices and hemodynamics variables, respectively, to compare the responses of all variables to both RE protocols. If the ANOVA was deemed significant, a Bonferroni correction factor was used for all post hoc testing. Paired t-tests were used for plasma volume changes and RPE to determine the difference between the two RE protocols. Partial eta-squared ($\eta p^2$) was used to assess the effect size (ES) and repeated measures ANOVA was used to determine statistical power ($1 - \beta$). If Mauchley's test of Sphericity was violated a Greenhouse-Geiser adjustment was used accordingly. A p-value of $P < 0.05$ was considered as statistically significance. All data are presented as mean±SD.

## Results

### Platelets indices and fibrinogen

Statistical analysis indicated no significant RE protocol x time interactions ($P > 0.05$) for platelet indices, fibrinogen, PDW, or plasma volume changes. Nonetheless there was a significant main effect of time for MPV ($F_{2,20} = 22.0$, $P = 0.0001$, ES = 0.68, power = 1.0) and P_LCR ($F_{2,20} = 19.8$, $P = 0.0001$, ES = 0.66, power = 1.0; Table 3). There were no significant differences in MPV and P_LCR in response to either RE protocol ($P > 0.05$), MPV and P_LCR were significantly decreased after 60 minutes of recovery in both RE protocols compared to baseline ($P < 0.05$; (Table 3). There were also no main effects of time for PLT (($F_{2,20} = 5.0$, $P = 0.05$, ES = 0.61, power = 0.39), fibrinogen ($F_{1.1,9.5} = 1.03$, $P = 0.35$, ES = 0.11, power = 0.15), PDW ($F_{2,20} = 3.5$, $P = 0.05$, ES = 0.26, power = 0.59) (Table 3), or plasma volume changes (Fig 1).

### Hemodynamic variables and RPE

There were significant interactions for HR ($F_{13,78} = 6.06$, $P = 0.001$, ES = 0.50, power = 1.0) and RPP ($F_{13,91} = 1.8$, $P = 0.049$, ES = 0.20, power = 0.87). HR and RPP were increased significantly in response to both RE protocols with both HR and RPP being increased in response to the IRE protocol when compared with the TRE protocol ($P < 0.05$) (Fig 2A and 2B) at IP. In addition to being elevated at IP compared to baseline, HR remained augmented up to 20 minutes into recovery compared to baseline for both RE protocols. There were also no significant interactions ($P > 0.05$) for SBP, DBP, or MAP. However, a significant main effect of time was

**Table 3. Mean values (±SD) of platelet indices, fibrinogen, and RPE at baseline and after (post) and recovery (After 60 min) in both resistance exercise protocols (N = 12).**

| | IRE | | | TRE | | |
|---|---|---|---|---|---|---|
| | Baseline | Post | Recovery | Baseline | Post | Recovery |
| PLT ($10^9$/l) | 250±50 | 274±52 | 248±41 | 251±39 | 256±40 | 249±37 |
| Men | 245±35 | 263±33 | 236±29 | 249±41 | 259±43 | 251±37 |
| women | 255±68 | 286±71 | 261±52 | 251±41 | 251±39 | 245±39 |
| MPV (fl) | 10.0±0.4 | 10.0±0.3 | 9.8±0.4* | 9.9±0.3 | 9.9±0.4 | 9.7±0.3* |
| Men | 10±0.4 | 10±0.3 | 9.9±.3 | 9.9±0.4 | 9.9±.04 | 9.7±0.3 |
| Women | 10±0.5 | 9.9±0.4 | 9.7±0.2 | 9.8±0.2 | 9.9±0.3 | 9.7±0.2 |
| P_LCR (%) | 26.1±3.2 | 26.0±2.9 | 24.6±3.4* | 24.8±2.6 | 25.3±2.7 | 23.9±2.3* |
| Men | 26±2.7 | 26±3.0 | 25±2.7 | 25±3.5 | 25±3.2 | 23±2.5 |
| Women | 26±4.0 | 25±2.9 | 24±4.4 | 24±1.4 | 25±2.1 | 24±2.1 |
| PDW (%) | 13±0.9 | 13±0.9 | 12±1.0 | 13±0.9 | 13±0.5 | 12±0.7 |
| Men | 13±0.7 | 13±1.1 | 12±1.0 | 13±1.0 | 13±0.6 | 12±0.8 |
| Women | 13±1.1 | 13±0.7 | 12±1.2 | 13±0.8 | 13±0.1 | 12±0.5 |
| Fib (mg/dL) | 251±43 | 265±78 | 239±49 | 243±65 | 246±65 | 226±49 |
| Men | 263±34 | 296±91 | 243±43 | 256±61 | 264±82 | 249±50 |
| Women | 236±53 | 228±41 | 235±60 | 227±73 | 225±31 | 191±19 |
| RPE | | 13±1.6 | | | 13±2.3 | |
| Men | | 12±1.3 | | | 13±1.8 | |
| Women | | 14±1.4 | | | 13±3.0 | |

PLT, Platelet; MPV, Mean Platelet Volume; PDW, Platelet Distribution Width; P_LCR, Platelet Large Cell Ratio Fib, Fibrinogen; RPE, Rate Perceived Exertion; IRE, Interval Resistance Exercise; TRE, Traditional Resistance Exercise.

*indicates different from baseline (P < 0.05).

observed for SBP (F13,91 = 8.02, P = 0.0001, ES = 0.53, power = 1.0) such that SBP increased significantly in response to both RE protocols IP (Fig 3A). In addition, SBP showed a significant reduction during recovery five minutes after RE compared to IP (P < 0.05) (Fig 3A). There were no significant changes (P > 0.05) in DBP or MAP in response to the RE protocols (Fig 3B and 3C). RPE was also not significantly between the RE protocols (Table 1).

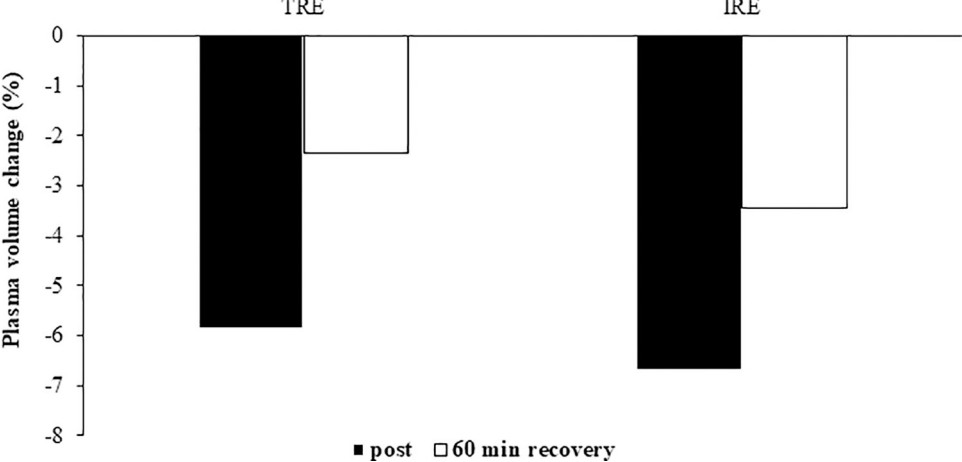

**Fig 1. Plasma volume levels immediately after both resistance exercise protocols and 60 minutes after them.** Values are means ± SD. IRE, Interval Resistance Exercise; TRE, Traditional Resistance Exercise.

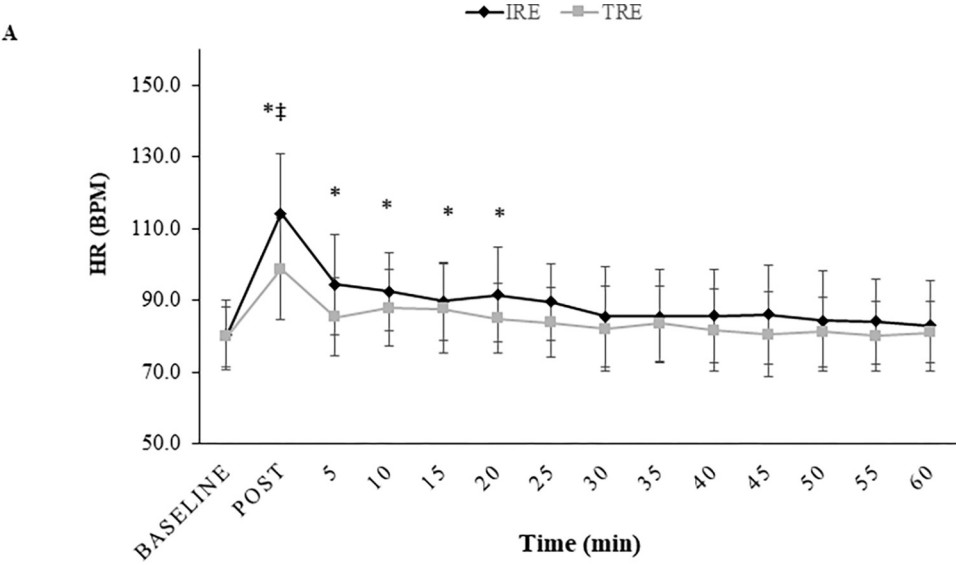

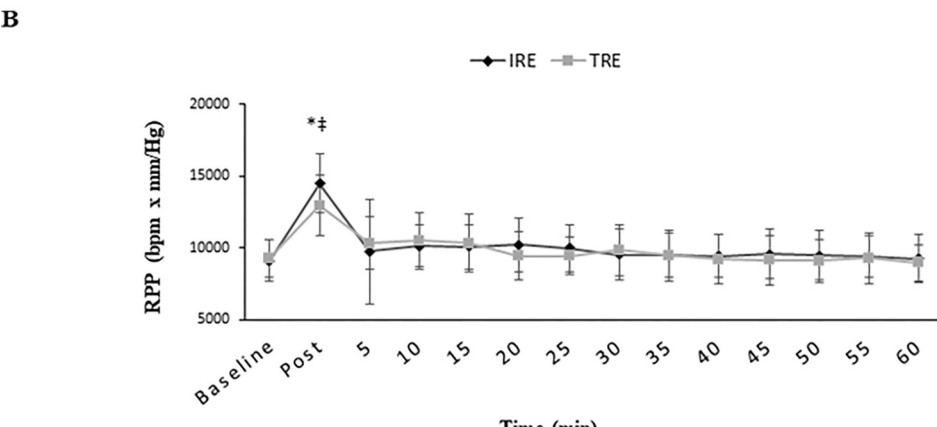

**Fig 2. Heart rate and Rate Pressure Product at baseline and immediately after resistance exercise protocols and every five minutes after them until 60 minutes.** (a) **HR**, Heart Rate; (b) **RPP**, Rate Pressure Product; **IRE**, Interval Resistance Exercise; **TRE**, Traditional Resistance Exercise. Values are means ± SD. * indicates a significant difference from baseline ($P < 0.05$); ‡ indicates a significant difference from traditional resistance exercise ($P < 0.05$).

## Discussion

The major findings of the current study were i) there were no significant changes in PLT, fibrinogen, PDW, DBP, or MAP, following either of the two RE protocols, ii) SBP, HR and RPP increase immediately in response to both RE protocols, iii) MPV, and P_LCR were significantly reduced during recovery compared to baseline. These data suggest that both RE protocols provide minor alterations in platelet indices and hemodynamics and reinforce that RE is a safe modality in young, healthy individuals.

In the present study, while our data demonstrated that neither RE protocol significantly altered PLT, there were increases in PLT at IP following both RE protocols. Following the TRE protocol PLT increased by 2.0%, but following the IRE protocol there was a 9.6% increase. This was partially in agreement with our hypothesis and supports our previous study, which

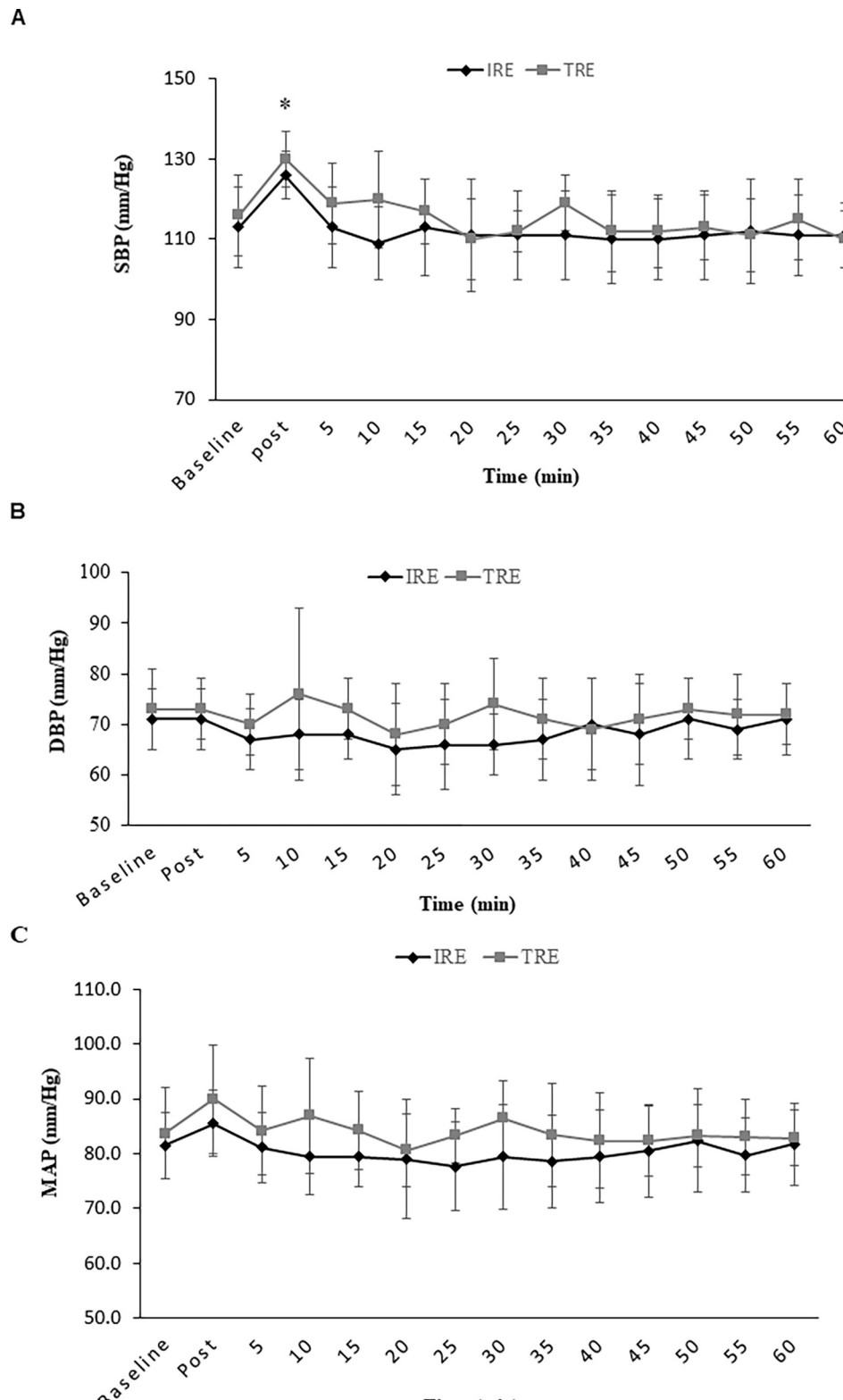

**Fig 3. Blood pressure at baseline and immediately after resistance exercise protocols and every five minutes after them until 60 minutes.** (a) **SBP**, Systolic Blood Pressure; (b) **DBP**, Diastolic Blood Pressure; (c) **MAP**, Mean Atrial Pressure; **IRE**, Interval Resistance Exercise; **TRE**, Traditional Resistance Exercise Values are means ± SD. *indicates a significant difference from baseline (P < 0.05).

indicated the different intensities of RE (40%, 60%, and 80%1RM) have similar effects on the PLT response [16]. It has been shown that exercise induces catecholamine release in an intensity-related manner, and the intensity of exercise has a positive effect on the stimulation of catecholamines [24,25]. An increase in sympathetic activity and catecholamine concentration immediately following exercise stimulates spleen contraction, and thus releasing young platelets from the spleen leads to an increase in the PLT response [25–27]. Furthermore, other sources for platelets like bone marrow and lungs could also contribute to this increase [28,29]. However, this increase in the PLT response was not different between the two RE protocols. We postulated because of the similar mean intensity (volume and intensity were equalized), both protocols would lead to similar physiological stress and consequently have an equal effect on stimulation of catecholamine.

In partial agreement with our hypothesis the current study indicated no change in MPV and P_LCR immediately-post RE. However, these markers showed significant reductions after 60 minutes of recovery compared to immediately-post RE. Both MPV and P_LCR are related to platelet size [30], and it has been shown that they have thrombogenic potential effects, and their increase could be associated with the development of various diseases like cardiovascular disease, hypertension, and thrombotic complications in those individuals with atherosclerosis [31,32]. An increase in MPV and P_LCR following RE may be related to the release of young and large platelets from the spleen by catecholamine stimulation, similar to the PLT response [26]. Interestingly, our results showed MPV, and P_LCR, significantly decreased and returned to baseline levels during 1-hr recovery following both protocols. Moreover, in this study, although fibrinogen concentration increased following IRE and TRE protocols (5.1% and 2.1%, respectively) this increase was not significant which disagreed with our hypothesis. There is a relationship between platelet form and inflammation status; thus, elevation in inflammation could induce platelet activity and therefore increase the likelihood of thrombotic events [33,34]. Thus, no significant increases in MPV, P_LCR or fibrinogen concentration in response to either RE protocol demonstrates that the RE protocols are safe and are not associated with adverse cardiovascular events.

This study demonstrated that SBP, HR, and RPP significantly increased in response to both RE protocols, and interestingly, these increases in HR and RPP were significantly greater in the IRE protocol immediately-post RE compared to immediately following the TRE protocol. These results confirmed our original hypothesis. However, in contrast to our hypothesis, in both RE protocols, SBP, HR, and RPP returned to baseline values during recovery. SBP and RPP returned to baseline by five minutes and HR recovered by 20 minutes. Previous studies have shown that RE has a significant role in increasing BP and HR and our study supports these previously reported results [15,35]. This increase in SBP and HR in response to RE might be attributed to stimulation of central command and the autonomic nervous system, which is associated with decreased parasympathetic activity, increased sympathetic activity, leading to an overall increase in HR and peripheral vascular resistance [36–38]. Furthermore, mechanical muscle reflex and accumulation of metabolic muscle products could have regulatory effects on autonomic modulation allowing for increases in SBP and HR [36,38]. Surprisingly despite no difference in the response of SBP between the two RE protocols, HR and RPP increased significantly greater following the IRE protocol immediately after the protocol compared to the TRT protocol. It is well established that during recovery time between exercises HR decreases because of a decrease in mechanical and metabolic reflexes [39–41]. We postulated that active recovery between exercises in the IRE protocol inhibited the restoration of autonomic (mechanical and metabolic stimulation) modulation and, as a result, would lead to a higher HR and RPP during IRE session. On the other hand, our data showed a decrease in SBP, HR, and RPP during 60 minutes recovery following RE. We assumed these decreases in both

platelet indices and hemodynamic variables, and also the non-significant changes in fibrinogen and PDW, in response to both RE protocols, might indicate that both RE protocols are safe in healthy and normotensive individuals, however further studies are needed to confirm these results, particularly in hypertension patients or people with cardiovascular diseases.

In the present study, our results indicated that RPE was similar between IRE and TRE protocols (13.3 and 13.4, respectively). It has been shown that RPE has a direct relationship with the autonomic nervous system and also catecholamine, lactate, and cortisol concentrations [36,37,42,43]. Our data are not surprising as both protocols were equalized (volume and intensity) to elicit similar psychological stress and perceived exertion.

Our study is associated with some limitations. The first one was a small sample size. Precise conclusions for a clinical intervention are needed, with adequate subjects, and thus future studies are warranted to be conducted with larger sample sizes in order to get precise results the effects of RE intensity and recovery on platelet indices and hemodynamic variables. Additionally, a more frequent measurement of platelet indices could have offered a more detailed picture of the recovery pattern of these variables. Furthermore, our study group was composed of both men and women. While our statistical analysis did not indicate any significant differences between genders in the study variables, the small sample size in each group may limit the statistical power to definitively conclude the absence of gender differences. Moreover, our study was conducted on healthy young adults with the lowest risk for cardiovascular events to determine the risk associated with intensity and recovery type between RE exercises; future studies must be performed on older adults and people with cardiovascular complications.

## Conclusion

Our study indicated that both RE protocols, irrespective of their intensity and recovery types, induced elevations that were transient, and both platelet indices (MPV, and P_LCR) and hemodynamic variables (SBP, HR, and RPP) returned to baseline levels within 1-hr of recovery. On the other hand, HR and RPP showed a greater increase in response to IRE protocol than TRE protocol. Based on these results, we consider that both intensity and recovery type during RE are influential variables on the hemodynamic response, and RE, irrespective of its and recovery types between exercises, could be safe in healthy individuals. However, future studies must confirm these results in high-risk people like hypertension and cardiovascular patients.

## Supporting information

**S1 File. Supporting information 1 (Hemodynamic variables data) SBP, Systolic Blood Pressure; DBP, Diastolic Blood Pressure; Mean Atrial Pressure; HR, Heart Rate; RPP, Rate Pressure Product; RPE, Rate Perceived Exertion; IRE, Interval Resistance Exercise; TRE, Traditional Resistance Exercise.**
(XLSX)

**S2 File. Supporting information 2 (Platelets indices data) PLT, Platelet; MPV, Mean Platelet Volume; PDW, Platelet Distribution Width; P_LCR, Platelet Large Cell Ratio Fib, Fibrinogen; REC, recovery; IRE, Interval Resistance Exercise; TRE, Traditional Resistance Exercise.**
(DOCX)

**S3 File. Supporting information 3 PLOS' questionnaire on inclusivity in global research.**
(XLSX)

## Acknowledgments

We would like to thank all participants who generously contributed to this study. No funds were received for this study.

## Author Contributions

**Conceptualization:** Mohammad Soltani.

**Data curation:** Mohammad Soltani, Fatemeh Hoseinzadeh.

**Formal analysis:** Mohammad Soltani, J. Derek Kingsley.

**Investigation:** Atefe Sarvestan, Fatemeh Hoseinzadeh.

**Methodology:** Mohammad Soltani, Atefe Sarvestan.

**Project administration:** Sajad Ahmadizad.

**Supervision:** Sajad Ahmadizad.

**Validation:** Mohammad Soltani.

**Writing – original draft:** Mohammad Soltani.

**Writing – review & editing:** Mohammad Soltani, J. Derek Kingsley.

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
