## [Decision Letter · Decision Letter 0]

14 Jul 2023

PONE-D-23-16264The Effects of Type of Recovery in Resistance Exercise on Responses of Platelet Indices and Hemodynamic VariablesPLOS ONE

Dear Dr. Soltani,

Thank you for submitting your manuscript to PLOS ONE. After careful consideration, we feel that it has merit but does not fully meet PLOS ONE’s publication criteria as it currently stands. Therefore, we invite you to submit a revised version of the manuscript that addresses the points raised during the review process.

We look forward to receiving your revised manuscript.

Kind regards,

Jianhong Zhou

Staff Editor

PLOS ONE

Reviewers' comments:

Reviewer's Responses to Questions

**Comments to the Author**

1. Is the manuscript technically sound, and do the data support the conclusions?

Reviewer #1: Yes

Reviewer #2: Yes

2. Has the statistical analysis been performed appropriately and rigorously? 

Reviewer #1: Yes

Reviewer #2: Yes

3. Have the authors made all data underlying the findings in their manuscript fully available?

Reviewer #1: Yes

Reviewer #2: Yes

4. Is the manuscript presented in an intelligible fashion and written in standard English?

Reviewer #1: Yes

Reviewer #2: Yes

5. Review Comments to the Author

Reviewer #1: Minor comments

1- In line 30, it seems “and exercises” is a typo.

2- The exercise protocols in the abstract could written more briefly.

3- Reporting the main or interactive effects of statistical analyses in the abstract do not make sense in clarifying the exact differences. please only mention the results of final analysis.

4- Please recheck the phrase “hyperplatelet activity,”

5- In line 139, it is mentioned that “and the third phase consisted of variables taken immediately after RE up to 60 min”, while only HR and BP were measured and not all the study variables.

6- In line 140, the same measurements up to 60 min mentioned in line 139 was repeated which makes the readers confused.

7- Fibrinogen is absent in the introduction.

8- While the exact reason for measuring HR and BPs and RPP every 5 min up to 60 min post-acute resistance exercise was not mentioned in the introduction, the absence of blood sampling during this period, limits to make any link between possible cardiac events with platelets or fibrinogen and etc. please mention to this topic in study limitations.

9- In line 257, the P-LCR reduction at 1 h recovery, is repeated in line 258.

Major comments:

1- The introduction has mentioned that exercise and possibly resistance exercise could increase the risk of cardiovascular events via changing platelet indices and hemodynamic variables. Firstly, no evidence was cited linking acute resistance exercise to cardiac event in healthy population. Secondly, the possible mechanisms that could hypothesize different effects from active versus passive recovery on the study variables was not indicated in more detail.

2-

ΔPV(%) was calculated using

ΔPV(%) = 100 × ((Hbpre-Hbpost) × (100 − Htc) − 1)

However, the correct formula should be as the following.

ΔPV (%) = 100 × ((HBpre/HBpost) × (100−HTCpost)/(100−HTCpre) −1)

Moreover, the parameter correction equation was not mentioned in the text. (To correct measured parameters for changes in PV, the following calculation should be used: [parameter]c = [parameter] u*(1 + ΔPV(%)/100), where the c and u indices represent corrected and uncorrected concentrations, respectively). Therefore, it is even possible the whole data and text could be renewed.

Reviewer #2: Dear Authors

I have reviewed the manuscript and found it to be well-designed. However, I believe that some corrections are necessary to improve the quality of the paper. Could you please provide your corrections and explanations, using the numbers I have included in my review.

1. Regarding the topic of the manuscript, your main goal was to investigate the effect of the type of recovery on platelet indices, why did you not clearly state the result of this investigation in the discussion section of the abstract, like what you did in the first paragraph of your discussion, but in paraphrased form.

2. It is suggested to mention the type of your subjects in the abstract. It will provide readers with a clear understanding of the population being studied and the relevance of your study to its main population.

3. Page 9/lines 59-61: It is not clear. It needs to be rewritten.

4. Considering having male and female subjects in the research, how did you control the effect of gender on the results? As is not mentioned anywhere in the manuscript.

5. Since the data of all variables are reported in a non-disaggregated manner, it is suggested to report non-disaggregated data in addition to separate data for men and women in Table 1.

6. Table 2/ lines: 154-156. Was the two-minute rest between exercises also active in IRE? If yes, how? With respect to the previous or the next exercise?

7. Page 15/ what does the phrase in brackets mean? A different symbol is used for eta squared.

8. In the discussion, considering that your subjects were healthy and that fibrinogen and MPV (like other variables) did not change significantly as expected, how do you explain the necessity of conducting your research?

6. PLOS authors have the option to publish the peer review history of their article (what does this mean?). If published, this will include your full peer review and any attached files.

Reviewer #1: **Yes: **Dr Karim Azali Alamdari

Reviewer #2: No

---

## [Author Response · Author response to Decision Letter 0]

27 Jul 2023

We are sincerely grateful to the reviewers for their insightful feedback, which we believe has significantly enhanced the readability and quality of our manuscript. We appreciate the time and effort they dedicated to provide such comprehensive and thoughtful reviews.

We have carefully addressed the reviewers’ comments and suggestions, and incorporated revisions accordingly. These changes are marked in the manuscript using the 'track changes' feature for ease of identification.

Response to Reviewer 1 (Minor comments):

1- In line 30, it seems “and exercises” is a typo.

Author response: Thank you for the comments. Changed now. P.3., L30.

2- The exercise protocols in the abstract could written more briefly.

Author response: Thank you for your insightful suggestion to condense the exercise protocols within the abstract. We have carefully considered your advice. However, given the complexity of our study, we believe the level of detail in the current 243-word abstract is necessary to maintain clarity and accuracy. We genuinely appreciate your understanding as we strive for precision in our work.

3-Reporting the main or interactive effects of statistical analyses in the abstract do not make sense in clarifying the exact differences. Please only mention the results of final analysis.

Author response: Changed now. P.3., L.34 to L.40.

4- Please recheck the phrase “hyperplatelet activity,

Author response: Thank you for your comment regarding the term "hyperplatelet activity". This term is indeed used in the scientific literature to describe elevated levels of platelet activation or reactivity. Its use in our manuscript aligns with its established context in the field. We appreciate your meticulous attention to detail and your understanding.

5- In line 139, it is mentioned that “and the third phase consisted of variables taken immediately after RE up to 60 min”, while only HR and BP were measured and not all the study variables.

Author response: We value your acute attention to detail. The information has now been expanded upon and revised for clarity. P.7., L.141 to L.142. 

6- In line 140, the same measurements up to 60 min mentioned in line 139 was repeated which makes the readers confused.

Author response: We appreciate your thorough and detailed review. The information has now been expanded upon and revised for clarity. P.7., L.141 to L.142.

7- Fibrinogen is absent in the introduction.

Author response: Thank you for your great point. Now we added fibrinogen in the introduction. P.4., L.60 to 62. 

8- While the exact reason for measuring HR and BPs and RPP every 5 min up to 60 min post-acute resistance exercise was not mentioned in the introduction, the absence of blood sampling during this period, limits to make any link between possible cardiac events with platelets or fibrinogen and etc. please mention to this topic in study limitations.

Author response: The selection of 60 minutes with 5-minute intervals was based on the dynamic nature of hemodynamic variables. Such interval monitoring, as reflected in prior studies, allows for more precise surveillance following exercise and captures time-related trends effectively.

In terms of the platelet indices recovery time, we grounded our decision in existing literature, which suggests that a longer duration is typically required for these indices to recuperate. However, we fully acknowledge your point and concur that this is indeed a limitation of our study.

As you rightly pointed out, our approach might restrict our ability to capture certain changes that occur outside these time intervals. We appreciate this astute observation, and we have now included this point in the limitations section of our paper. P.16., L. 331 to L. 333.

9- In line 257, the P-LCR reduction at 1 h recovery, is repeated in line 258.

Author response: Thanks, we now deleted the repeated sentence. P.12., L259 to 260. 

Response to reviewer 1:

Major comments 

 1- The introduction has mentioned that exercise and possibly resistance exercise could increase the risk of cardiovascular events via changing platelet indices and hemodynamic variables. Firstly, no evidence was cited linking acute resistance exercise to cardiac event in healthy population. Secondly, the possible mechanisms that could hypothesize different effects from active versus passive recovery on the study variables was not indicated in more detail.

Thank you for your insightful query regarding the link between acute resistance exercise and the risk of cardiac events in a healthy population.

Author response: In the literature review conducted by the American Heart Association and published in Circulation, it was highlighted that even in healthy individuals; vigorous exercise could elevate the risk of cardiac events, with resistance exercise being labeled as vigorous. “Vigorous exercise transiently increases the risk of AMI and SCD, even in exercise-conditioned individuals”[1]. Furthermore, additional studies over the past decade have shown vigorous exercise could increase the risk of cardiac events [2, 3]. However, the reviewer is correct. Direct links between resistance exercise and cardiac events are minimal in the literature. But, based on previous research, the effects of resistance exercise on the cardiovascular are robust and have alluded to negative effects. Nonetheless, more data and ongoing research are needed to affirm this conclusion.

As we mentioned in our manuscript, prior research has demonstrated that resistance exercise augments platelet activation, as evidenced by an upsurge in platelet aggregation and a rise in beta-thromboglobulin (B-TG) levels [4, 5]. Consequently, these observations suggest that vigorous exercise, including resistance exercise, could possibly increase the risk of cardiac events, particularly immediately after the exercise session.

We understand the importance of clarity and rigorous referencing in our work. Therefore, we appreciate your critical observation and will ensure to include more explicit evidence in the revision.

Thank you for your great comments regarding the need for elaboration on the potential mechanisms differentiating the effects of active versus passive recovery on the study variables.

We appreciate your keen observation and have now expanded it to include a more detailed explanation of the physiological mechanisms we hypothesize could account for the differences between passive and active recovery (P.4, L85 to 92). Your feedback has significantly enhanced the comprehensiveness of our study.

2) 

ΔPV(%) was calculated using

ΔPV(%) = 100 × ((Hbpre-Hbpost) × (100 − Htc) − 1)

However, the correct formula should be as the following.

ΔPV (%) = 100 × ((HBpre/HBpost) × (100−HTCpost)/(100−HTCpre) −1)

Moreover, the parameter correction equation was not mentioned in the text. (To correct measured parameters for changes in PV, the following calculation should be used: [parameter]c = [parameter] u*(1 + ΔPV(%)/100), where the c and u indices represent corrected and uncorrected concentrations, respectively). Therefore, it is even possible the whole data and text could be renewed.

Author response: Thank you for your keen observation regarding the plasma volume calculation. You are correct; the formula presented in our original manuscript was not accurate. This was due to an unfortunate typographical error. However, we indeed used the correct formula, as you correctly mentioned in your comments. Following your helpful feedback, we have now made the necessary correction in the revised manuscript. (P.9., L.184.)

Thank you for your meticulous review and the insightful suggestion concerning the potential use of a parameter correction equation according to plasma volume for our variables. As you astutely noted, such a correction could be important under certain circumstances. However, as presented in our manuscript, the analysis revealed no significant difference in plasma volume between conditions in our study. Given that the plasma volume remained stable throughout the study, it suggests that the use of a correction formula did not alter our results. While we appreciate the importance of such corrections in situations of significant fluid balance alterations, the lack of significant change in plasma volume in our study lends us to believe that our findings are representative without the application of a correction formula.

Response to Reviewer 2:

1. Regarding the topic of the manuscript, your main goal was to investigate the effect of the type of recovery on platelet indices, why did you not clearly state the result of this investigation in the discussion section of the abstract, like what you did in the first paragraph of your discussion, but in paraphrased form.

Author response: Thanks for your brilliant comment, we have changed it now. P.2., L41 to 42.

2. It is suggested to mention the type of your subjects in the abstract. It will provide readers with a clear understanding of the population being studied and the relevance of your study to its main population.

Author response: Thanks. It has been added. (P. 2., L.27.).

3. Page 9/lines 59-61: It is not clear. It needs to be rewritten.

Author response: Thanks for the comments. We rewrote it. P.3. L.58 to L.63.

4.Considering having male and female subjects in the research, how did you control the effect of gender on the results? As is not mentioned anywhere in the manuscript.

Author response: We sincerely appreciate your astute question and insightful comments regarding the impact of gender on our research outcomes. To address your concern we added some detail about female participants and their menstrual cycle. P.5., L.108 to 110.

For our female participants, we meticulously planned our tests to coincide with the first five days of the early follicular phase of their menstrual cycle, a period associated with the lowest levels of oestrogen which helped us control for potential hormonal influences.

To ensure the comparability of results across genders, we carried out a comprehensive statistical analysis to identify potential differences in the response to exercise between men and women. Our analysis revealed no significant differences in this regard.

However, we recognize, as you rightly pointed out, that our small sample size (six subjects of each gender) limits the effect size and statistical power concerning sex differences. This is indeed a significant limitation of our study. Acknowledging this, we have now incorporated this point into the limitations section of our manuscript. P.16., L.333 to L.336.

Your thoughtful question not only deepens our understanding but also helps readers more fully comprehend our research and its constraints. We extend our profound gratitude for your perceptive contribution.

5. Since the data of all variables are reported in a non-disaggregated manner, it is suggested to report non-disaggregated data in addition to separate data for men and women in Table 1. 

Author response: Thank you for your thoughtful suggestion. We have added separate data for both men and women in Table 1. P.11. to 12. 

6. Table 2/ lines: 154-156. Was the two-minute rest between exercises also active in IRE? If yes, how? With respect to the previous or the next exercise?

Author response: Thank you the great comment. During the IRE protocol, we included an active rest phase between exercises, where participants continued with 20% 1RM. After finishing the three sets of a particular exercise, participants were given a two-minute inactive rest period before starting a new exercise. We have added further explanation in the manuscript to ensure the methodology is comprehensively understood. P.7., L.150 to L.152.

7. Page 15/ what does the phrase in brackets mean? A different symbol is used for eta squared.

Author response: Thank you for your meticulous attention to detail. The term in question was indeed a typographical error. The correct term is partial eta-squared (ηp²). We have now made the necessary correction. P.9., L.206.

8. In the discussion, considering that your subjects were healthy and that fibrinogen and MPV (like other variables) did not change significantly as expected, how do you explain the necessity of conducting your research?

Author response: We appreciate your valuable comments. Contrary to our original hypothesis, we did not observe any significant changes in platelet indices and fibrinogen, which was unexpected. We believe that if a larger sample size was utilized, we might have detected significant variations, particularly in fibrinogen levels. 

Further, our study assumed that due to the elevated physiological stress associated with the IRE protocol, there might be a disruption in our measured variables or an extended recovery time. This hypothesis formed the basis of our current research.

Previous studies have suggested that heightened physiological stress, induced by increased intensity, can negatively impact hemodynamic variables and platelet indices. However, our investigation aimed to explore whether alterations in recovery type could amplify physiological stress, consequently affecting our study variables.

References:

1. Medicine, A.C.o.S., et al., Exercise and acute cardiovascular events: placing the risks into perspective: a scientific statement from the American Heart Association Council on Nutrition, Physical Activity, and Metabolism and the Council on Clinical Cardiology. Circulation, 2007. 115(17): p. 2358-2368.

2. Stuckey, M., et al., Autonomic recovery following sprint interval exercise. Scandinavian journal of medicine & science in sports, 2012. 22(6): p. 756-763.

3. Miron, V.V., et al., High-intensity intermittent exercise increases adenosine hydrolysis in platelets and lymphocytes and promotes platelet aggregation in futsal athletes. Platelets, 2019. 30(7): p. 878-885.

4. Ahmadizad, S. and M.S. El-Sayed, The effects of graded resistance exercise on platelet aggregation and activation. Medicine and science in sports and exercise, 2003. 35(6): p. 1026-1032.

5. Ahmadizad, S., M.S. El-Sayed, and D.P. MacLaren, Effects of time of day and acute resistance exercise on platelet activation and function. Clinical hemorheology and microcirculation, 2010. 45(2-4): p. 391-399.

---

## [Decision Letter · Decision Letter 1]

2 Aug 2023

The Effects of Type of Recovery in Resistance Exercise on Responses of Platelet Indices and Hemodynamic Variables

PONE-D-23-16264R1

Dear Dr. Soltani,

We’re pleased to inform you that your manuscript has been judged scientifically suitable for publication and will be formally accepted for publication once it meets all outstanding technical requirements.

Kind regards,

Daniel Boullosa

Academic Editor

PLOS ONE

Additional Editor Comments (optional):

Reviewers' comments:

Reviewer's Responses to Questions

**Comments to the Author**

1. If the authors have adequately addressed your comments raised in a previous round of review and you feel that this manuscript is now acceptable for publication, you may indicate that here to bypass the “Comments to the Author” section, enter your conflict of interest statement in the “Confidential to Editor” section, and submit your "Accept" recommendation.

Reviewer #1: All comments have been addressed

Reviewer #2: All comments have been addressed

2. Is the manuscript technically sound, and do the data support the conclusions?

Reviewer #1: Yes

Reviewer #2: Yes

3. Has the statistical analysis been performed appropriately and rigorously? 

Reviewer #1: Yes

Reviewer #2: Yes

4. Have the authors made all data underlying the findings in their manuscript fully available?

Reviewer #1: Yes

Reviewer #2: Yes

5. Is the manuscript presented in an intelligible fashion and written in standard English?

Reviewer #1: Yes

Reviewer #2: Yes

6. Review Comments to the Author

Reviewer #1: (No Response)

Reviewer #2: (No Response)

7. PLOS authors have the option to publish the peer review history of their article (what does this mean?). If published, this will include your full peer review and any attached files.

Reviewer #1: **Yes: **Azali Alamdari, Karim

Reviewer #2: No

---

## [Editor Report · Acceptance letter]

8 Aug 2023

PONE-D-23-16264R1 

The Effects of Type of Recovery in Resistance Exercise on Responses of Platelet Indices and Hemodynamic Variables 

Dear Dr. Soltani:

I'm pleased to inform you that your manuscript has been deemed suitable for publication in PLOS ONE. Congratulations! Your manuscript is now with our production department. 

Kind regards, 

on behalf of

Dr. Daniel Boullosa 

Academic Editor

PLOS ONE